# AN EXPLORATION OF LEARNT REPRESENTATIONS OF W JETS

**Jack H. Collins**
SLAC National Accelerator Laboratory
`jcollins@slac.stanford.edu`

## ABSTRACT

I present a Variational Autoencoder (VAE) trained on collider physics data (specifically boosted $W$ jets), with reconstruction error given by an approximation to the Earth Movers Distance (EMD) between input and output jets. This VAE learns a concrete representation of the data manifold, with semantically meaningful and interpretable latent space directions which are hierarchically organized in terms of their relation to physical EMD scales in the underlying physical generative process. The variation of the latent space structure with a resolution hyperparameter provides insight into scale dependent structure of the dataset and its information complexity. I introduce two measures of the dimensionality of the learnt representation that are calculated from this scaling.

## 1 INTRODUCTION

Energetic events at the Large Hadron Collider (LHC) consist of hundreds of particles each described by four momentum components, leading to embedding spaces with dimensionality $\mathcal{O}(1000)$. Dimensionality reduction is therefore important for understanding this data. Distance measures between events based on optimal transport such as Earth Movers Distance (EMD) have been introduced for usage on particle physics datasets in recent years (Komiske et al., 2019b; 2020a; Crispim Romão et al., 2021; Komiske et al., 2020b; Cai et al., 2020), leading to geometric interpretations of the data manifold from which many useful quantities can be derived. Different generative processes involved in creating events are associated with distinct EMD scales.

Variational Autoencoders (VAEs) (Kingma & Welling, 2014) have been shown to produce semantically meaningful and interpretable dimensional reductions into their latent space in many contexts. To be trained, they require a notion of similarity between pairs of objects to be used in the reconstruction loss. Pixel intensity based losses have been used in various VAE studies for collider data (Dillon et al., 2021; Cheng et al., 2020; Dohi, 2020), but these fail to reflect the similarity of events with collinear splittings or small displacements. Given its appealing physical properties, and the physical interpretations of its corresponding manifolds, the EMD between events is a promising candidate to be used as a reconstruction loss in a VAE which might then be used for studying the data manifold in this space. This paper serves to introduce a VAE trained with such a reconstruction loss, and to describe some experiments on a dataset of $W$-jets, which are collimated streams of particles formed from the decay of a $W$ boson travelling with high momentum. Code used for generating the results for this paper can be found at `https://github.com/jackhcollins/EMD_VAE/tree/ICLR_DGM4HSD`.

## 2 EXPERIMENTS

**Data and Architecture**  A sample of $6 \times 10^5$ $W$ jets were simulated with transverse momenta in the range $500 - 600$ GeV using a standard pipeline described in Appendix A.1. The 50 largest-momentum particles are selected and stored as 3-vectors $(p_T, \eta, \phi)$. The momenta $p_T$ are prescaled so that their sum is one for each jet. Two VAEs with identical architecture are trained: `VAE_uncent` (`VAE_cent`) is trained on jets which have not (have) been centered in the detector. The VAEs are built with a Particle Flow Network (Komiske et al., 2019a) encoder which takes input jets as point clouds $\boldsymbol{x}$, and a dense decoder which outputs jets $\rho$ with the same structure. The encoder

parameterizes means and variances $\boldsymbol{\mu}(\boldsymbol{x})$, $\boldsymbol{\sigma}(\boldsymbol{x})^2$ for a multivariate diagonal normal distribution with 256 dimensions, from which latent space coordinates $\boldsymbol{z}$ are sampled. The loss function,

$$L_{\text{EMD}-\text{VAE}} = \frac{\hat{S}(x, \rho(z))^2}{2\hat{\beta}^2} + \sum_i \frac{1}{2} \left( \mu_i(x)^2 + \sigma_i(x)^2 - \log\left(\sigma_i(x)^2\right) - 1 \right), \tag{1}$$

is composed of a distortion ($D$) term $\hat{S}^2$, which is chosen to be a sharp Sinkhorn (Sinkhorn & Knopp, 1967; Cuturi, 2013; Schmitzer, 2016) approximation to the EMD, and a rate term ($R$) given by the expectation value of $D_{\text{KL}}(p(\boldsymbol{z}|\boldsymbol{x})\|\mathcal{N}(\boldsymbol{z};\mathbf{0},\boldsymbol{I}))$ which can be decomposed into a sum of separate contributions from each latent direction. $\hat{S}$ is dimensionless due to the $H_T \equiv \sum p_T$ rescaling of the jets, but it is related to true Sinkhorn distance between the physical jets by $S \equiv H_T \hat{S}$. Similarly, while the hyperparameter $\hat{\beta}$ is dimensionless, it is related to a dimensionful quantity defined as $\beta \equiv \langle H_T \rangle \hat{\beta}$. The dimensionality of the latent space is chosen to be larger than the full dimensionality of the dataset (150), so that an identity map is in principle possible and the information capacity is primarily restricted by the rate constraint rather than an architectural bottleneck.

**Training** $\beta$ can be interpreted equivalently as the noise parameter of the Gaussian posterior probability from which the reconstruction loss descends, setting the resolution scale of the VAE, or as the $\beta$ hyperparameter of a $\beta$-VAE (Higgins et al., 2017). The VAE is trained by $\beta$-annealing (Fu et al., 2019) whereby it is trained in stages using a sequence of values for $\hat{\beta}$, preserving the model weights between stages. This sequence is log-uniform separated in the range $10^{-5}-1$. The procedure begins with an initial 'priming' run, starting at the smallest $\hat{\beta}$ and proceeding upwards. Next, $\hat{\beta}$ is annealed in a zig-zag pattern until it reaches again its smallest value. Finally, a 'production' run is performed, repeating the sequence of values used for the priming run. The model weights saved at the end of each $\beta$ step in this run are used to generate the results of this paper.

Before proceeding to describe the learnt representations, I will make some qualitative observations from training. During most of the priming run the learnt representation is disorganized, taking advantage of all 256 latent directions to describe the data (i.e. all dimensions have associated $D_{\text{KL}}$ significantly greater than zero). By the production run the learnt representation has been organized into a small set of informative directions with $D_{\text{KL}} > 0$, while the majority are uninformative with $D_{\text{KL}} \simeq 0$. For each informative direction, there is some critical value $\hat{\beta}_{\text{crit}}$ above which the dimension becomes uninformative. These are associated with physical scales in the training data, for instance the translation of jets around the detector ($\pi H_T$), or the orientation of hard prongs within the jet ($m_W$). When trained with $\hat{\beta} \gg \hat{\beta}_{\text{crit}}$, the corresponding variations will not be learnt in the latent space. When trained with $\hat{\beta} \ll \hat{\beta}_{\text{crit}}$, the variations may be learnt in the latent space, but they have no tendency to be organized into orthogonal or semantically meaningful ones. When subsequently trained with $\hat{\beta} \lesssim \hat{\beta}_{\text{crit}}$, these dimensions tend to organize into a small number of semantically meaningful directions, which have a tendency to be preserved if $\hat{\beta}$ is subsequently gradually reduced to very small values. Further details of the training procedure are given in Appendix A.1, and plots of the $D_{\text{KL}}$ associated with the individual latent directions can be found in Fig. 7.

**Learnt representations** VAE_uncent has three informative directions at $\beta \simeq 100$ GeV, and an additional three at $\beta \simeq 10$ GeV. VAE_cent has no informative directions at $\beta \simeq 100$ GeV, three at $\beta \simeq 10$ GeV, and many more at smaller values of $\hat{\beta}$. For each VAE, the latent directions can be ordered by the sizes of their individual contributions to $R$. Fig. (1) illustrates the physics encoded by the first three $z_i$ of VAE_uncent with $\beta \simeq 100$ GeV. The blue contours which indicate the density of the training data in the latent space suggest that a two dimensional manifold has been embedded into a three dimensional space. $z_0$ maps $\eta$ while $z_1$ maps $-\pi/2 < \phi < \pi/2$. The remaining half of the barrel is mapped with the aid of $z_2$ which appears to encode $\text{sign}(\cos\phi)$, and a full $2\pi$ rotation around the detector barrel is obtained by traversing the ring in the $(z_1, z_2)$ plane.

Fig. (2) illustrates the representation learnt in the two most informative directions of VAE_cent at $\beta = 6$ GeV. The coordinates $(z_0, z_1)$ of this VAE map the polar and azimuthal angles $(\theta, \phi)$ of the $W$ decay in its rest frame, where $\theta$ can be mapped to the energy fraction $z$ of the hardest prong in the boosted frame. The topology of the decay of a massive particle into two identical particles is that of the real projective plane, $\text{RP}^2$, which in this context is naturally represented by the sphere with antipodes identified. Any hemisphere gives a single cover of the space of two-body decays

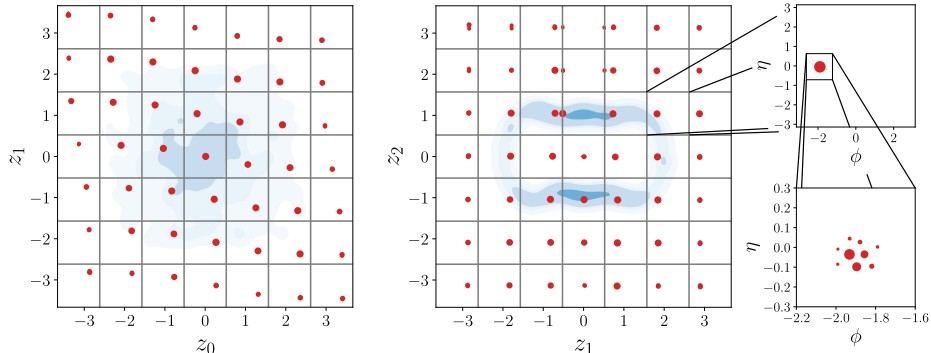

Figure 1: Learnt representation of jet coordinates for `VAE_uncent` with $\beta = 50$ GeV. The major axes of the left and center plots are the latent space coordinates $z_0$ to $z_2$. The blue contours indicate probability density of encoded events. Overlaid are grids of jet images in red, in which areas of discs are proportional to the $p_T$ of the corresponding particle. Each jet image has its own coordinate axes $(\phi, \eta)$, and is generated by decoding the latent code associated with the major axis coordinates of the center of its small square. The jet images on the right zoom in on one of these small squares.

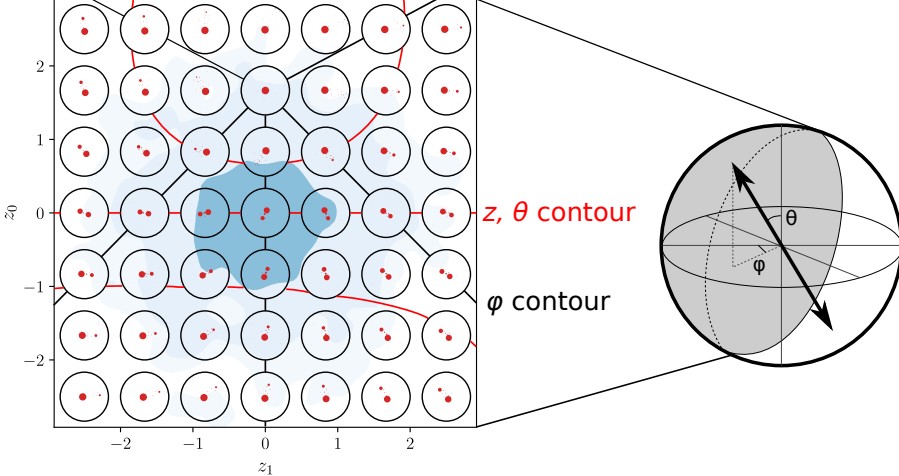

Figure 2: Learnt representation of two-body jet substructure in `VAE_cent` with $\beta = 6$ GeV. The major axes of the plot correspond to the latent directions $z_0, z_1$. Each circle contains a jet image with internal axes $(\phi, \eta)$ ranging from $-0.5$ to $0.5$. Each jet image is obtained by the decoder from the latent code associated with the coordinates of the center of the circle. Black and red lines are approximate contours of the polar and azimuthal angles $\theta$, $\phi$ of the $W$ boson decay, drawn by eye.

everywhere except on its rim, with an identification of opposite points on the rim being the surviving remnant of the antipodal identification on the full sphere. This plane of the latent space represents an approximate projection of the hemisphere indicated by the grey region on the right of the figure, with the pole of the sphere being represented at approximately $(0, 0.75)$. Jets located around the edge of the support region satisfy an approximate reflection symmetry $(z_0, z_1) \rightarrow (-z_0, -z_1)$. Additional latent dimensions are illustrated in Fig. (8).

**Dimensionality** The organization of the learnt information into a small number of latent dimensions that vary smoothly with $\beta$ suggests the possibility of notions of information complexity that depend on the way that properties of the learnt representation scale with $\beta$. To this end, I introduce definitions for two notions of dimensionality

$$D_1 = -\frac{dR}{d\log\beta}, \qquad D_2 = \frac{dD}{d\beta^2}, \tag{2}$$

where in practice these derivatives are estimated using finite difference approximations using quantities evaluated using VAEs trained at nearby values of $\beta$. These quantities should be equal along the optimal frontier since $2\beta^2 = -\partial R/\partial D$ (Alemi et al., 2018), and can be regarded as an analogue of a thermodynamic heat capacity (Alemi & Fischer, 2018). $D_2$ can be interpreted as a dimensionality by noting that if sampling in $D_2$ informative and orthogonal latent directions maps via the decoder to a stochastic sampling in $D_2$ orthogonal dimensions in the reconstruction space, then the full reconstruction error can be obtained by adding in quadrature those associated with the individual orthogonal directions. This leads to $D \simeq D_2\beta^2 + \text{const}$ (since $\beta$ is behaving as a gaussian noise parameter), and the derivative extracts the dimension. It is related to the work of Rezende & Viola (2018), in which $dD_2/d\beta^2$ is studied and spikes in this quantity are interpreted as indicating phase transitions. The interpretation of $D_1$ as a dimensionality stems from the qualitative observation that informative latent space directions scale like $\sigma_i \propto \beta$ (Fig 7), while uninformative ones have $\sigma_i \simeq 1$. The posterior $p(z|x)$ occupies a Gaussian ball in the latent space with volume $\text{Vol} \sim \prod \sigma_i \sim \beta^{D_1}$, with $D_{\text{KL}} \simeq \log(\prod \sigma_i)$. It can therefore be seen that $D_1$ plays the role of an exponent that relates a scale ($\beta$) to a quanitity which approximates a volume ($R = \langle D_{\text{KL}} \rangle$). In Appendix A.3 I describe a simple analytic example in which these formulae for $D_{1,2}$ can be derived exactly.

Fig. (3) plots the dimensions calculated on both VAEs during the production run. $D_1$ and $D_2$ are in rough agreement for both VAEs, except for at low values of $\beta$ for `VAE_uncent` where it struggles to represent the detailed substructure of the jet at the same time as its bulk position in the detector. At high scales, the dimensionality of `VAE_uncent` is very close to 2, while the corresponding physics is described using three latent dimensions as was illustrated in Fig. (1). The third latent dimension, acting as a categorical variable, contributes very little to the computed dimensionality. This illustrates the role that these quantities have in reflecting the true information complexity of the dataset when compared to a naive counting of active latent directions. Indeed, the dimensionality scaling of the learnt representation agrees with an intuitive understanding of the dataset. At large $\beta$, the uncentered jets have dimensionality of 2 while centered have dimensionality of 0. An order of magnitude below and three new dimensions emerge, two of which describe the orientation of the hard prongs within the jet associ-

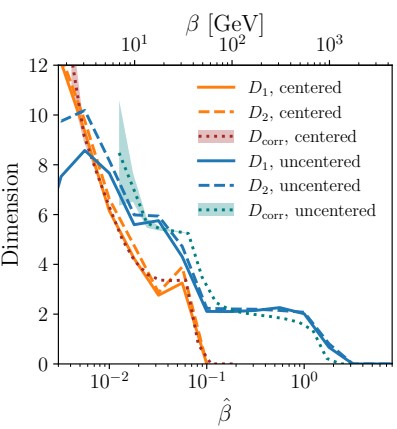

Figure 3: Representation dimension.

ated with the scale $m_W$ and a third one describing the overall boost of the jet (associated with the spread of momenta $\Delta H_T \simeq 100$ GeV). At scales below this many new dimensions rapidly emerge, representing the various physical processes associated with showering, hadronization, decays, and detector effects as discussed in Komiske et al. (2019b) and illustrated in Fig. (8). Also plotted are the correlation dimensions of the datasets which measure their scale-dependent complexity, defined by (Kégl, 2002)

$$D_{\text{corr}}(Q) = \frac{d}{d \log Q} \log \sum_{i,j \neq i} \Theta(Q^2 - \hat{S}^2(x_i, x_j)). \tag{3}$$

This was calculating also using the method of finite differences using a subset of $10^4$ events $\boldsymbol{x}$ from the training set, and is presented with statistical uncertainty bands. No direct relation between $Q$ and $\beta$ has been established, and so $D_{\text{corr}}(Q)$ is arbitrarily plotted with $Q = 2\beta$ which results in qualitative alignment with $D_{1,2}(\beta)$. While this work does not demonstrate a concrete link between $D_{\text{corr}}(Q)$ and $D_{1,2}(\beta)$, their qualitative similarity is intriguing.

## 3 CONCLUSIONS AND OUTLOOK

The VAEs trained for this study are effectively learning a concrete representation of the metric space of jets induced by the EMD. They identify semantically meaningful, intuitive, and approximately orthogonal principal axes of variation in the space of jets. Associated with these principal axes are concrete scales, which reflect scales associated with the physical generative process for the jets. As the resolution of the VAE is varied by adjusting $\beta$, the learnt representation smoothly adjusts, and

its varying properties can be probed to understand the information complexity of the EMD manifold of the jets. There remains to be seen potential applications for these properties, and the question of mixed samples which are expected in unsupervised training on LHC data.

ACKNOWLEDGMENTS

This work was supported by the U.S. Department of Energy, Office of Science under contract DE-AC02-76SF00515. This work was initiated at the Aspen Center for Physics, which is supported by National Science Foundation grant PHY-1607611. I am grateful for the support of staff maintaining the SLAC Shared Data Facility, on which the models were trained. I am also thankful to the maintainers of the software packages Matplotlib Hunter (2007), Jupyter Kluyver et al. (2016), NumPy Harris et al. (2020), and SciPy Virtanen et al. (2020). I would like to thank the following for many helpful discussions and continuing support, without which this work would not have been possible: Ben Nachman, Matthew Schwartz, Patrick Komiske III, Eric Metodiev, Zhen Liu, Jesse Thaler, Siddharth Mishra-Sharma, Michael Kagan, and Alex Alemi.

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

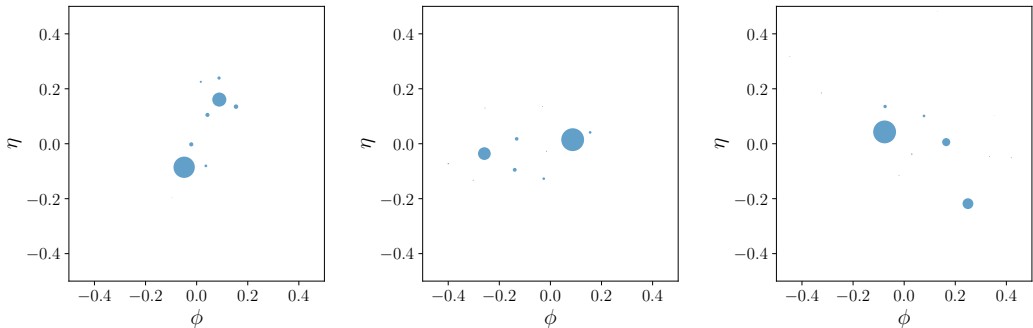

Figure 4: Training examples.

## A APPENDIX

### A.1 MORE DETAILS ON DATA, ARCHITECTURE, AND TRAINING

#### A.1.1 DATA DESCRIPTION FOR NON DOMAIN SPECIALISTS

The data used for this study (both training and testing) is simulated data from proton-proton collisions that occur at the Large Hadron Collider. In particular, a special subcategory of this data is selected, because it has a relatively simple and well-understood structure that is interesting at multiple physical scales. The data is coming from the decays of simulated $W$-bosons, travelling at relativisitic velocities and decaying into pairs of quarks. As the quarks separate, additional particles are produced around them typically with much lower momentum, following the theory of Quantum Chromodynamics. The end result of this process is the production of typically 10-100 particles, mostly clustered around two primary centers of activity (which represent the directions of the two quarks from the initial decay), although additional centers frequently develop.

Particles can each be desribed by their momentum 3-vector (if their mass is neglected, as it is in this study). This can be conveniently represented in polar coordinates $(E, \theta, \phi)$ around the proton-proton collision point, or equivalently (as is often conventional for hadron colliders) $(p_T, \eta, \phi)$ where $p_T = E \sin \theta$, and $\eta = -\ln(\tan(\theta/2))$, with $\eta \in (-5, 5)$ and $\phi \in [-\pi, \pi)$. Because of the relativistic velocity of the $W$, these particles tend to be collimated in the direction of the velocity vector of the $W$ boson, with a typical spread in the $\eta$-$\phi$ plane of around 0.3 for the chosen simulation parameters.

Fig. (4) shows three examples of $W$-jets, which have been centered to have their overall velocity vector to be in the direciton $(\eta, \phi) = (0, 0)$. Jets elsewhere in the detector can be obtained simply by translations of these in the $\eta$-$\phi$ plane. In order to produce these images particles that are very close (within a separation of around 0.02) are reclustered into single particles, just to aid visualization. Each particle is represented by a disc in the $(\eta, \phi)$, centered at the coordinates of the particle, and with area proportional to the $p_T$ of the particle. Clusters of energetic particles are represented by clusters of large discs. In the left and center image are shown two predominantly two-pronged examples with differing orienatations. The right image shows an example where a prominent third prong as emerged during the showering process.

Figure (5) gives a rough schematic of the generative processes. Translations of $W$-jets in the $(\eta, \phi)$ plane are associated with the production process and correspond to EMDs of $\sim H_T \Delta R$, where $\Delta R = \sqrt{\Delta \eta^2 + \Delta \phi^2}$ and $H_T \simeq 500 \, \text{GeV}$. The decay process determines the rotations of the two major prongs and their relative balance, variations of which correspond with EMDs of order $m_W = 80 \, \text{GeV}$. The showering and hadronization processes that result in the spread of particles around and between the two main centers of energy, and occasionally resulting in a third priminent prong, typically account for EMDs of up to $10 \, \text{GeV}$. The VAE tends to easily disentangle latent directions associated with these different generative processes, because they are associated with hierarchically separated scales. Similarly, the various intrinsic dimenions of the data manifold that are associated with these processes can be seen to emerge a little below their relevant scales in Fig. 3.

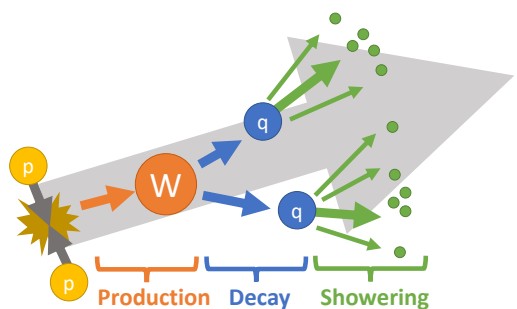

Figure 5: Schematic of generative processes.

### A.1.2 DATA GENERATION

$W$ jets are simulated and decayed in Madgraph (Alwall et al., 2011) with the process $pp \rightarrow WZ \rightarrow \nu\nu jj$ and a generation level cut on the missing momentum of $p_{T,\text{miss}} > 500$ GeV. The events are showered in Pythia8 (Sjöstrand et al., 2015). Detector simulation is performed with Delphes (de Favereau et al., 2014) using an ATLAS based card and with particle flow reconstruction. Particle flow objects were then clustered into jets with the anti-$k_t$ algorithm (Cacciari et al., 2008) with $R = 1$. The event is selected if the leading jet has momentum in the range $500 - 600$ GeV, $|\eta| < 2$, and mass in the range $75 - 110$ GeV. For selected events, the constituents of the leading jet are reclustered with anti-$k_t$ with $R = 0.07$ and the leading 50 particles are recorded with $(p_T, \eta, \phi)$. Events with fewer than 50 particles are zero padded. $6 \times 10^5$ events survive the cuts, of which $5 \times 10^5$ are used for training and the remainder for validation and testing.

### A.1.3 NETWORK AND TRAINING DETAILS

The inputs to the VAE are jets each with 50 particles represented as $\{(p_T/H_T, \eta, \sin\phi, \cos\phi)\}$. The encoder network of the VAEs consists of four 1D convolution layers with filter size 1024, kernel size 1, stride 1, followed by a sum layer, followed by four dense layers of size 1024. Unless otherwise specified, all layers have activation function Leaky ReLu with negative slope coefficient of 0.1. 256 latent space $\mu, \log \sigma^2$ are encoded with linear activation. The decoder consists of five layers with size 1024, followed by a linear dense layer which outputs fifty particles represented as $\{(p_T/H_T, \eta, \sin\phi, \cos\phi)\}$, and then an $\arctan$ function reduces this to $\{(p_T/H_T, \eta, \phi)\}$. The explicit $\arctan$ allows the network to avoid learning a discontinuity in $\phi$, which is also the motivation for the trigonometric form of the inputs.

The loss function is a custom implementation in TensorFlow of a sharp Sinkhorn (Luise et al., 2018) using $\epsilon$-scaling (Schmitzer, 2016; Sharify et al., 2013; Kosowsky & Yuille, 1994). In principle symbolic differentiation should be effective for this problem, however I encountered debilitating numerical instabilities that I was unable to diagnose. I therefore implemented the explicit gradient introduced in Luise et al. (2018). The Sinkhorn distance is calculated with regulator scaling from 1 to 0.01 in ten log-uniform steps, with ten iterations per step. Double floating precision is required for this calculation.

The $\hat{\beta}$-annealing schedule used for training `VAE_uncent` is illustrated in Fig. (6), each black dot representing a step in the annealing schedule. At each step, the VAE is trained for fifty epochs, or until validation loss has not improved for ten epochs. Training is performed with batch size of 100 and with 1000 steps per epoch, and so five epochs are required to cycle through the whole training dataset. The adam optimizer (Kingma & Ba, 2014) is used for training. The learning rate begins at $3 \times 10^{-5}$ at the beginning of each annealing step, and reduces by a factor of $\sqrt{0.1}$ if validation loss has not improved in five epochs. The first straight line of annealing steps is in this paper called the 'priming' run. The last straight line is called the 'production' run.

Training was performed on NVIDIA GeForce 2080Ti GPUs. An epoch takes approximately two minutes, and the full annealing schedule approximately four days.

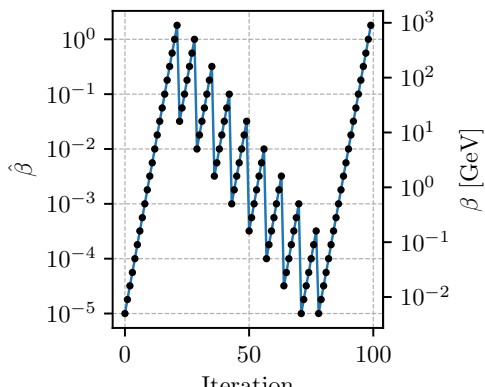

Figure 6: Annealing Schedule.

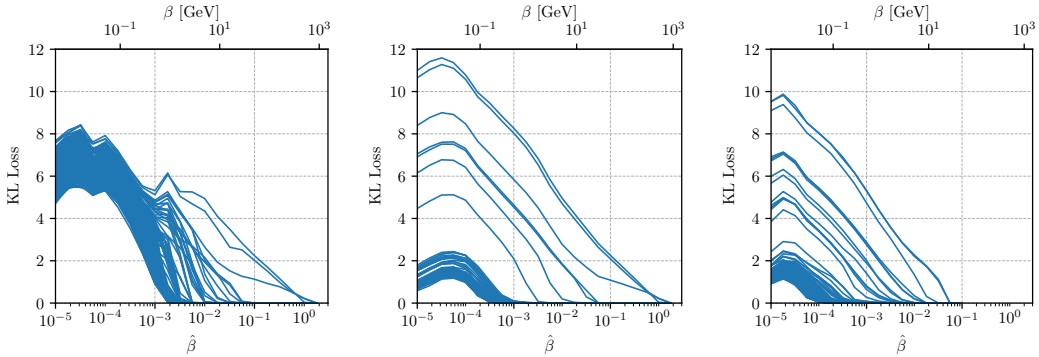

Figure 7: **Left:** Evolution of the individual KL losses associated with the 256 latent dimensions in the first annealing run for VAE_uncent. **Center:** Evolution of the individual KL losses associated with the 256 latent dimensions in the final annealing run for VAE_uncent. **Right:** Evolution of the individual KL losses associated with the 256 latent dimensions in the final annealing run for VAE_cent.

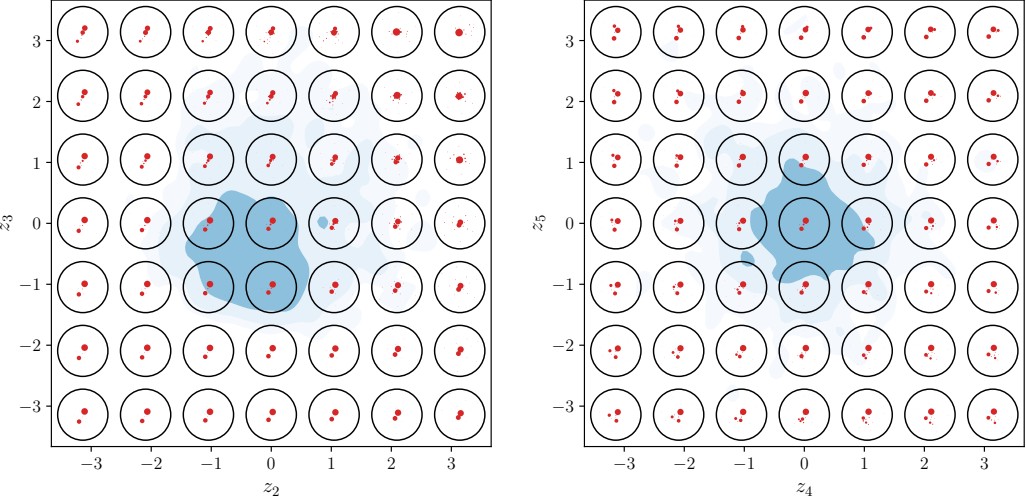

Figure 8: Additional latent dimensions of VAE_uncent. Refer to caption of Fig. (1) for details. **Left:** Visualization of latent space directions $z_2$, which appears to describe the boost of the jet, and $z_3$, which appears to describe the prominence of a third prong. **Right:** Visualization of latent space directions $z_4$ and $z_5$, which appear to describe the orientation of a third prong.

## A.2   ADDITIONAL LATENT DIRECTIONS

In Fig (8) is plotted slices of the latent space of VAE_uncent, in the directions $z_2$ to $z_5$. $z_2$ is very close to $z_0$ and $z_1$ in prominence (see Fig. (7), right), and describes the overall boost of the jet. Since the jets are generated with a $p_T$ range of 100 GeV, this corresponds to a EMD variation that is comparable to those associated with the decay process. $z_3$ to $z_5$ describe the first three latent variables associated with the showering process. $z_3$ appears to determine the prominence of a third prong, while the $(z_4, z_5)$ plane determines its relative orientation. Moving radially out from the origin in this plane moves the third prong away from the center of the jet, while traversing a circle in this plane moves the angle of the third prong with respect to the dominant two.

## A.3   A SIMPLE ANALYTICAL EXAMPLE

Consider a toy one-dimensional gaussian distributed dataset with variance $\bar{\sigma}$. A linear VAE with one latent dimension is trained to reconstruct the coordinate of the input $x$, with loss function

$$L_{\text{toy VAE}} = \frac{(x - \rho(z))^2}{2\beta^2} + \frac{1}{2} \left( \mu(x)^2 + \sigma(x)^2 - \log\left(\sigma(x)^2\right) - 1 \right). \tag{4}$$

The VAE is linear in the sense that $\mu(x)$, $\sigma(x)$, $\rho(z)$ are linear functions of their arguments. The loss function can be integrated analytically over $p(x)$ and $p(z|\mu, \sigma)$, and has extrema given by

$$\mu(x) = \rho(z) = 0, \quad \sigma(x) = 1, \tag{5}$$

$$\mu(x) = \pm \frac{\sqrt{\bar{\sigma}^2 - \beta^2}}{\bar{\sigma}^2} x, \quad \rho(z) = \pm\sqrt{\bar{\sigma}^2 - \beta^2}\, z, \quad \sigma(x) = \beta/\bar{\sigma}. \tag{6}$$

The former results in an uninformative latent space with $D_{\text{KL}} = 0$ which is a minimum only for $\beta > \bar{\sigma}$, and becomes a saddle point for $\beta < \bar{\sigma}$. The latter, which leads to an informative latent space, is real only for $\beta < \bar{\sigma}$ and is a minimum in this regime. Substituting this minimum into the expressions for the reconstruction and KL losses in Eq. (4) gives

$$D = \left\langle (x - \rho(z))^2 \right\rangle_{p(x)p(z|x)} = \beta^2 \tag{7}$$

$$R = \left\langle D_{\text{KL}} \right\rangle_{p(x)} = -\log\left(\beta/\bar{\sigma}\right). \tag{8}$$

Evaluating Eq. (2) explicitly gives $D_1, D_2 = 1$ for $\beta < \bar{\sigma}$ and $D_1, D_2 = 0$ for $\beta > \bar{\sigma}$.

The general case of a $d$-dimensional Gaussian dataset with variances $\{\bar{\sigma}_i\}$ is more complicated, but it can be shown that there are minima when $d$ latent space axes are aligned with the principal axes of the data. In these minima, the problem reduces to $d$ independent copies of the one-dimensional case. In this case, $D_1, D_2$ both count the number of directions for which $\beta < \bar{\sigma}_i$, which is the same as the number of active dimensions that have $D_{\mathrm{KL}} > 0$

$$D_1, D_2 = \sum_i \Theta\left(\bar{\sigma}_i - \beta\right). \tag{9}$$

In summary, the VAE probes in detail directions that have characteristic scale larger than $\beta$, and averages over directions which have scale smaller than $\beta$. The $\bar{\sigma}_i$s play the role of the $\beta_{\mathrm{crit}}$s of Section 2.

Nonlinear VAEs trained on non-Gaussian data have no guarantee to follow this behaviour, in which case the behaviour of these quantities can be regarded as a diagnostic of how closely the behaviour of the data resembles that of a Gaussian.

### A.4 RELATION TO PREVIOUS MACHINE LEARNING WORK

Earth Movers Distance has been commonly used as a reconstruction error for generative models of unweighted point clouds starting with Fan et al. (2017); Achlioptas et al. (2018). In the case of unweighted point clouds this is often implemented using an auction algorithm, but this is unsuitable for weighted point clouds as in this work. Instead, this work closely follows the approach of Patrini et al. (2020), which introduced Sinkhorn autoencoders for image datasets. The implementation of this work differs from that of Patrini et al. (2020) by applying the Sinkhorn distance to weighted point clouds rather than images. Essentially this just means that the locations of the pixels are allowed to vary, in addition to their intensity. It is a relatively trivial change with additional gradients for the locations of the points, but to my knowledge it is novel, as I am not aware of any previous attempt in the literature to build an autoencoder with an EMD-based loss for weighted point clouds.

