# OpenReview forum: "An Exploration of Learnt Representations of W Jets"
_ICLR.cc/2022/Workshop/DGM4HSD — ICLR 2022 DGM4HSD workshop Poster_

### Official Review · Reviewer_hSLh · 2022-03-21
**Concise physics-based application of VAEs**

**Rating:** 8
**Confidence:** 3

**Review:**

Note that I am not an expert on any of the physics within the paper.

### Quality

The paper is of great quality for a workshop submission. The problem and the approach to solve it are well-motivated and communicated clearly. The experiments are both thorough and convincing. The only thing missing is a related work section - I'm left wondering things like "how does this paper relate to Wasserstein Auto-encoders / Adversarial Variational Bayes / etc" but I understand the workshop format is quite limiting in what can be included in the paper.

### Clarity

As a non-expert in physics, the writing still feels fairly understandable to me, which I would like to give the author credit for. The method and experimental choices are well-explained in the work.

### Originality

I am not totally sold that the approach is entirely original - other works, as noted above, have considered how to use Wasserstein distance in the context of VAEs - but I have not seen this exact problem studied using VAEs in the literature (although, again, not an expert here).

### Significance

Again, I find it hard to comment on the significance of this paper to the physics community. To the machine learning community, the methods here are not completely groundbreaking, but it is nice to get further perspective on what types of methods work on real data, and not just examples such as MNIST and CIFAR-10.

### Pros and Cons

__Pros:__
- Interesting application to real data
- Thorough explanation of the approach taken, including hyperparameters and things like annealing schedule for $\beta$
- Thoughtful experiments

__Cons:__
- No placement within the larger generative modelling literature

---

### Official Review · Reviewer_dwVa · 2022-03-23
**Interesting application of VAEs to particle physics, few suggestions for improving clarity of the paper**

**Rating:** 7
**Confidence:** 3

**Review:**

In this work, the author introduces a VAE trained with EMD between events as a reconstruction loss to describe experiments on a dataset of W-jets. The paper is well-written and provides nice background on other methods that have been used in the author's field, and I have made some recommendations below to potentially improve the paper further. The code was not provided so I did not review that, although the author claims it will be made available after anonymous peer review.

While I cannot comment on the novelty/significance of the physics aspects of the work, I found this to be an interesting application of VAEs and it seems to be an appropriate method for this task as the latent space seems to provide interesting physical insights. The figures are also very nice.

Recommendations/comments:
* I understand there is perhaps not enough room with the page limit, but it would be nice for the author to elaborate more on the physics for out-of-field folks reading this paper (for instance, by also providing a few concrete examples of what the training data looks like as well as some visualizations).
* A schematic summarizing the model and the data going in and coming out of the decoder would be nice, including a summary of the latent space, to help readers understand what is being done.
* Unclear how much was done in terms of hyperparameter optimization, can the author comment on this?
* Would also be nice to see the validation loss curves, to get a sense of how meaningful the learned representations are for points outside the training set.

I recommend for the paper to be accepted although there are still a few things that could be done to improve the clarity.

---

### Official Review · Reviewer_MS3z · 2022-03-27
**Good work, but not sure if fitting this workshop**

**Rating:** 6
**Confidence:** 3

**Review:**

The authors present an empirical exploration of the VAE-learned representations for a high energy physics dataset of simulated W-jets. The methodological contribution of the paper is to use Earth Movers Distance as a distortion metric/unnormalized likelihood for the VAE used in this context. The issue with that contribution is that the authors do not investigate or present how that contribution affects the learned representations, or their utility, in comparison to other distortion metrics such as pixel intensity based losses mentioned by the authors. Although the problem setting is definitely a case of highly structured data, somehow the domain-specific and case-study (rather than methodological contribution) nature of this manuscript makes me think that this work belongs more to a workshop along the lines of ML4PhysicalSciences than this one. Because of these venue-fit and report-like-nature (I’m tempted to call it an exploratory analysis of the learned representations), I give it only “slightly above threshold” rating, and leave the decision to the discretion of the organizers.

Otherwise, the quality of the work is good: it’s well-written, very well produced (great figures!), it was an interesting and entertaining read, and I haven’t found any glaring technical errors. I found the domain-specific description in the last paragraph on page 2 a bit difficult to follow, but it was not crucial to understanding of the other parts of the paper.

As I said earlier, I think comparing the representations learned with different likelihoods could leave the reader with more of a take-home message (“EMD likelihood allows to learn representations corresponding to true generative factors faster/easier/better than pixel intensity likelihood”), because at the moment, in my opinion, the core weakness of the paper is that is lacks a clear conclusion/motivation.

---

### Decision · Program_Chairs · 2022-03-26

Accept (Poster)